# Bacterial Adhesion on Glass–Ionomer Cements and Micro/Nano Hybrid Composite Dental Surfaces

Klemen Bohinc [1,*], Erna Tintor [2], Davor Kovačević [2], Rajko Vidrih [3], Anamarija Zore [1], Anže Abram [4], Željka Kojić [5], Marija Obradović [5], Valentina Veselinović [5] and Olivera Dolić [5]

1 Faculty of Health Sciences, University of Ljubljana, 1000 Ljubljana, Slovenia; anamarija.zore@zf.uni-lj.si
2 Department of Chemistry, Faculty of Science, University of Zagreb, 10000 Zagreb, Croatia; erti2802@gmail.com (E.T.); davork@chem.pmf.hr (D.K.)
3 Biotechnical Faculty, University of Ljubljana, 1000 Ljubljana, Slovenia; rajko.vidrih@bf.uni-lj.si
4 Department for Nanostructured Materials, Jožef Stefan Institute, 1000 Ljubljana, Slovenia; anze.abram@ijs.si
5 Faculty of Medicine, University of Banja Luka, 78000 Banja Luka, Bosnia and Herzegovina; zeljka.kojic@med.unibl.org (Ž.K.); marija.obradovic@med.unibl.org (M.O.); valentina.veselinovic@med.unibl.org (V.V.); olivera.dolic@med.unibl.org (O.D.)
* Correspondence: klemen.bohinc@zf.uni-lj.si

**Abstract:** Dental restorations need to reproduce the aspect of the natural teeth of the patient, and must be non-toxic, biocompatible, and have good mechanical properties so that they can last for longer. The aim of this study was to determine the extent of bacterial adhesion of *Streptococcus mutans* on four different dental material surfaces, i.e., two glass–ionomer cements (Fuji conventional and Fuji hybrid) and two ceramic composites (Micro hybrid composite and Nano hybrid composite). To understand the bacterial adhesion on these four different dental materials, various surface properties were measured: roughness, contact angle, CIE color parameters and zeta potential. We found that the greatest adhesion extent was obtained for the Nano hybrid composite surface. The pronounced adhesion is the interplay between the relatively high roughness and hydrophilicity of the Nano hybrid composite surface. Color changes upon immersing ceramic composites in red wine and black tea proved that both beverages adhered to them. Black tea adhered more intensively than wine, and showed a higher inhibitory effect on the growth of *Streptococcus mutans* and *Staphylococcus aureus*.

**Keywords:** bacterial adhesion; glass–ionomer cement; Micro/Nano hybrid composite; surface properties





## 1. Introduction

Dental restorative materials are regarded as artificial predilection sites for the adherence and accumulation of oral microorganisms [1]. To prevent oral diseases (caries, gingivitis, periodontitis, peri-implantitis) dental materials with a low susceptibility to bacterial adhesion are preferable for the longevity of restorations [2,3]. A rough composite and glass–ionomer resin surface may increase bacterial biofilm accumulation. This may lead to an increased risk of caries and periodontal inflammation [4,5].

Brushing, polishing, abrasion, erosion and microcracking processes, as well as acid medium can modify composite and glass–ionomer dental restorative materials. A very important property of composites and glass–ionomers is hydrophobicity, which affects the initial water absorption and the adhesion of oral bacteria [6,7]. The contact angle method gives an average value for hydrophobicity; the measured angle is low when the surface is hydrophilic. Namen et al. [7] obtained high contact angles in dry conditions, especially in the case of finished and polished samples, regardless of the liquid used for measuring. Several other previous studies [8–10] showed that dental plaque formation is smaller on hydrophobic materials such as amalgams and resins than on hydrophilic restorative materials such as porcelain and metals. However, other studies [11–13] report greater plaque formation on hydrophobic materials and significantly lower adhesion to

ceramics than to composite resin surfaces of polymeric origin or amalgams. Additionally, it has been shown that the contact angle on a solid surface decreases as the surface becomes rougher [14,15].

The color of the teeth is determined by the combined effects of intrinsic and extrinsic colorations. Intrinsic tooth color is associated with the light scattering and absorption properties of the enamel and dentine [16]. Extrinsic color is associated with the absorption of materials (e.g., tea, red wine, chlorhexidine, iron salts) onto the surface of enamel, and the pellicle coating, and which ultimately causes extrinsic stain [17].

Color characteristics of teeth or restorative materials might be determined according to the Commission Internationale de l'Eclairage (CIE) parameters (*L**, *a**, *b**). *L** represents lightness (0 = yields black, 100 = indicates diffuse white); *a** negative values indicate green and positive values indicate red; and *b** negative values indicate blue and positive values indicate yellow. Additional calculations from basic (*L**, *a**, *b**) parameters are Chroma (difference from grey color) and hue angle, which encodes red, orange, yellow, green, blue, and purple. $\Delta E$ is another parameter which determines the color difference, taking into account all three basic CIE parameters.

In order to quantify the clinical significance of the differences between two colored samples, color difference thresholds have been introduced in dentistry. Paravina et al. [18] reported that values smaller than a color difference $\Delta E = 1.2$ are not perceptible, while values greater than $\Delta E = 2.7$ are clinically unacceptable.

Bacterial adhesion processes are affected by physico-chemical properties of the bacterial and material surfaces [19]. Physico-chemical properties are determined by environmental conditions, material surface properties and bacterial properties. The material surfaces are characterized by roughness, hydrophobicity, and charge, while the bacterial surface is characterized by hydrophobicity, charge, flagellation, and motility [20]. Theoretically, the bacterial adhesion is generally described by a two-stage binding model. First, a reversible interaction between the bacterial cell surface and the material surface takes place. The bacterial adhesion is governed by van der Waals, electrostatic, hydrophobic effects, acid–base pairs, and contact interactions [21,22]. In the simplest situation, the interaction Gibbs free energy of adhesion process shows two minima. The first minimum appears at the separation of a few tens of nanometers and is a few thermal energies deep. The microorganism is weakly and reversibly bound. The second minimum corresponds to the specific and nonspecific interactions between so-called adhesion proteins expressed on bacterial surface structures and binding molecules on the material surfaces. The interaction free energy appears at a contact distances of a few nanometers. This means that the microorganism is strongly and irreversibly adhered. The bacteria must surpass a large energy barrier of a few thermal energies to overcome from the first into the second minimum at the contact.

For our study, *Streptococcus mutans* (*S. mutans*) was especially important, because it is mainly responsible for caries development. In primary caries, the bacteria initiate lesions in virgin tooth structure. In secondary caries, the bacterial adhesion takes place in dental restoration, particularly at material margins. As already noted, material properties greatly affect bacterial adhesion. Some of them can exhibit better marginal fit, finish, and polish than direct materials [23–25].

The aim of this study was to determine the bacterial adhesion rate on four different dental material surfaces, i.e., two glass–ionomer cements (Fuji conventional and Fuji hybrid) and two ceramic composites (Micro hybrid composite and Nano hybrid composite). For improved understanding of the bacterial adhesion, the surface properties needed to be investigated. Therefore, surface roughness was determined by profilometry, hydrophobicity by contact angle measurements, and zeta potential by measuring the streaming potential. The bacterial adhesion rate was determined from scanning electron microscopy (SEM) micrographs. For the purpose of our study, *S. mutans*, which is the main etiological agent for caries formation and primary colonizing bacteria of the oral cavity, was chosen. For all materials, we also determined the color parameters and studied the change of color parameters by dipping the surfaces into typical red wine and black tea samples.

## 2. Materials and Methods

### 2.1. Bacteria and Growth Conditions

*Streptococcus mutans* (*S. mutans*) ATCC 25,175 strains used in this study were selected from culture on blood-agar plates incubated at 37 °C for 48 h with a $CO_2$ pack for anaerobic conditions. The *S. mutans* overnight culture was made in brain–heart infusion (BHI) nutrient broth (Biolife, Italiana Srl) (4012302) at 37 °C for 24 h to obtain a $10^9$ CFU/mL bacterial suspension. In this suspension, we incubated samples of different dental materials for 10 h, the attached bacteria were fixed, and the samples were examined with a scanning electron microscope. One milliliter of bacterial suspension (in cell concentration $10^9$ CFU mL$^{-1}$) from overnight culture was taken and diluted to a proximal cell concentration of $10^7$ CFU mL$^{-1}$ in a fresh nutrient broth (brain–heart infusion—BHI broth) and cultivated for 5, 10, 15, 20 and 24 h at 37 °C. The growth of bacteria was measured with spread plate counts on BHI agar. Eventually, the inhibitory effect of tested materials on bacterial growth was tested in parallel aliquots with the presence of tested materials (plates of 1 cm$^2$).

*Staphylococcus aureus* (*S. aureus*) was used as a hemolytic strain. This bacterium is Gram-positive, cocci shaped, and tends to aggregate into small clusters. *S. aureus* is normally found in human microbiota, on skin and in the nasal area, but if it enters the bloodstream it is a potential pathogen. It can cause multiple kinds of skin and pulmonary infections, as well as meningitis, gastroenteritis, and sepsis. Obtained *S. aureus* was incubated on blood agar at 37 °C for 16–24 h. After incubation, one-third of an inoculation loop of pure bacterial culture was transferred to 5 mL of BHI nutrient broth, vortexed, and incubated at the same conditions to obtain an overnight culture. The overnight culture was diluted 1:300, inoculated onto samples, and incubated at the same conditions.

The inhibitory effect of black tea and red wine on the bacterial growth was tested by the Kirby–Bauer disk diffusion method. The antimicrobial susceptibility test was performed in Mueller–Hinton agar.

### 2.2. Material Properties of Dental Surfaces

Two glass–ionomer cement materials were used in the study: Fuji conventional (Fuji IX) and Fuji hybrid (EQUIA Forte Fil) and two composite materials: Micro hybrid composite (TE Econom) and Nano hybrid composite (Tetric Evo Ceram) as presented in Table 1.

A total of 40 cylindrical test specimens of dimensions 18 mm × 0.2 mm (diameter × height) were prepared from a mold using Fuji IX, EQUIA Forte GIC, TE Econom and Tetric Evo Ceram composite; 10 of each group. The powder and liquid of the glass–ionomer cements (GIC) were mixed according to the manufacturer's instructions and placed in the molds. The EQUIA Forte capsules were mixed in the capsule mixer for 10 s, then removed and placed in the molds with the help of the GC capsule applier. The mixed cement was placed into the mold by slightly overfilling them and covering with the Mylar strips placed between mold and the glass plate to prevent the adhesion of GIC to the glass plate. The glass plates were held firmly during setting to avoid the presence of air bubble and to obtain a smooth surface. TE Econom and Tetric Evo Ceram composite was placed into the mold using a plastic instrument covered with a Mylar strip and cured with a light-activated source. A glass slide of 1–2 mm thick was placed over the strip before curing with the light-curing unit to flatten the surfaces.

**Table 1.** Description of materials used in this study.

| Material | Type | Manufacturer | Composition |
|---|---|---|---|
| Fuji conventional (Fuji IX) | Glass–ionomer cement | GC (Tokyo, Japan) | Powder: 95% by weight alumino-fluoro-silicate glass with 5% polyacrylic acid powder. Liquid: 50% distilled water, 40% polyacrylic acid, and 10% polybasic carboxylic acid. |
| Fuji hybrid (Fuji equia Forte glass–ionomer cements) | Glass–ionomer cement | GC (Tokyo, Japan) | Powder: 95% strontium fluoro alumino-silicate glass, 5% polyacrylic acid Liquid: 40% aqueous polyacrylic acid |
| Micro hybrid composite (TE Econom composite) | Micro hybrid composite | Ivoclar, Vivadent (Liechtenstein) | Matrix: Dimethacrylate and TEGMA (22 wt.%). Fillers: barium glass, ytterbium trifluoride, silicon dioxide and mixed oxide (76 wt.% or 60%vol) |
| Nano hybrid composite (Tetric EvoCeram composite) | Nano hybrid composite | Ivoclar, Vivadent (Liechtenstein) | bis-GMA, UDMA, ethoxylated bis-EMA, barium glass, ytterbium trifluoride, spherical mixed oxide, acyl phosphine oxide (75 wt.%) |

*2.3. Roughness Measurements*

For the characterization of the surface topography of dental surfaces, profilometry was used. From imaging data, the quantitative evaluation of surface features was performed. From the statistical analysis, the roughness parameters were determined. On each type of dental surfaces, three-line measurements in the length of 5 mm were made. From the profilometer data, the analysis of the roughness was made and the arithmetic average roughness ($R_a$), and root mean square roughness ($R_q$) were calculated.

Surface topology was characterized using a Form Talysurf Series 2 (Taylor-Hobson Ltd., Leicester, UK) profilometer with a resolution of 0.25 µm, 1 µm and 3 nm in the *x*, *y*, and *z* directions, respectively. A set of parallel line scans was performed with a tip of 2 µm. Data were processed using TalyGold, Taylor Hobson, Leicester, UK. To separate roughness from waviness, a Gaussian cut-off filter of 0.8 mm was used. The surfaces could thus be characterized regarding height, spatial, and hybrid parameters, as specified in ISO 25178.

*2.4. Contact Angle Measurements*

Contact angle measurements were made using an Attension Theta (Biolin Scientific, Gothenburg, Sweden) tensiometer, which consisted of a light source, camera, liquid dispenser, and a sample stage. Dental surfaces were placed on the sample stage. A liquid droplet was put on the material surface and the contact angle between the droplet and the surface was measured. Eight measurements were made for each material, from which the average value of the contact angle was calculated.

*2.5. Zeta Potential Measurements*

The surface charge analysis was accomplished using an electro-kinetic analyzer (SurPASS, Anton Paar GmbH, Graz, Austria). The streaming potential was obtained in a 1 mM phosphate-buffered saline (PBS) solution via streaming potential measurements at room temperature. From the measured streaming potential, the zeta potential $\zeta$ was calculated.

### 2.6. X-Ray Diffraction

The phase composition and crystallinity of the samples were analyzed using an X-ray diffractometer (XRD, PANalytical X'Pert PRO, Malvern Panalytical, Malvern, UK, $CuK_\alpha$ = 1.5406 Å).

### 2.7. Scanning Electron Microscopy with Energy Dispersice X-ray Spectroscopy

For standard observations, SEM (Jeol JSM-7600F, Jeol Ltd., Tokyo, Japan) with a thermal field-emission gun (FEG) and in-lens secondary electron detector (SEI), lower secondary electron detector (LEI), backscattered electron detector (BE), and energy-dispersive X-ray spectrometer (EDXS, X-MAX, Oxford Instruments, Oxfordshire, UK) was utilized. Electron micrographs were obtained using SEI, LEI, and BE detectors at accelerating voltages from 5–15 kV and working distances 2–15 mm.

### 2.8. Color Measurements

Surface color was measured with a colorimeter (CR-400; Minolta, Kyoto, Japan). Color parameters (*L\**, *a\**, *b\**) were analyzed according to the Commission Internationale de l'Eclairage (CIE). Additional color parameters, i.e., the hue angle (°), were calculated as $arctg(b^*/a^*)$ and the relative color saturation (*C\**) was calculated as $(a^{*2} + b^{*2})^{1/2}$. The hue angle is represented quantitatively by a single number, typically on a color wheel, where 0° represents red color, 180° its complementary green color, 90° represents yellow color, and 270° its complementary blue color. Relative color saturation (*C\**) represents colorfulness of an object as judged according to its brightness.

In this experiment, the change in color after dipping in red wine or black tea was checked by colorimetry. Plates were submerged in red wine or black tea for 10 min and allowed to dry after being removed from both beverages. The Commission Internationale de l'Eclairage (CIE) parameters *L\**, *a\** and *b\** were recorded before and after dipping the plates. The total color difference ($\Delta E$) was calculated as $\Delta E = ([\Delta a^*]^2 + [\Delta b^*]^2 + [\Delta L^*]^2)^{1/2}$, where $\Delta E > 2.7$ corresponds to "very distinct", $1.2 < \Delta E < 2.7$ to "distinct", and $\Delta E < 1.2$ to "non-distinct" changes.

### 2.9. Adherence of Black Tea and Red Wine on Dental Surface

Fuji conventional and Fuji hybrid dental plates were used to investigate the adherence of black tea and red wine on their surface. Both dental plates were immersed in black tea (standard infusion) and red wine for 10 min. After immersion, plates were removed and allowed to dry in air. The adherence of both beverages was monitored by measuring CIE *L\**, *a\**, *b\** color parameters (see Section 2.8) with respect to intact plates.

## 3. Results

### 3.1. Growth of Bacteria

Growth curves of bacteria *S. mutans* and *S. aureus* were measured. The peak in the bacterial growth of *S. mutans* was reached after 10 h incubation at 37 °C. The peak in the curve corresponded to approx. $0.9 \times 10^9$ CFU/mL of culture. Due to the lack of nutrients in the broth, the bacterial growth was suppressed after 10 h.

### 3.2. Roughness

The roughness of dental surfaces was measured using the profilometer with a stylus that ran on the surface of a sample. From the profilometer's data, the arithmetic average roughness $R_a$ and root mean square roughness $R_q$ were calculated and are presented for all four materials in Figure 1. The highest roughness was measured for the Nano hybrid composite: $R_a$ = (2 ± 0.2) μm and $R_q$ = (2.6 ± 0.4) μm. The other materials had slightly lower roughness. The lowest was for the Micro hybrid composite: $R_a$ = (1.4 ± 0.4) μm.

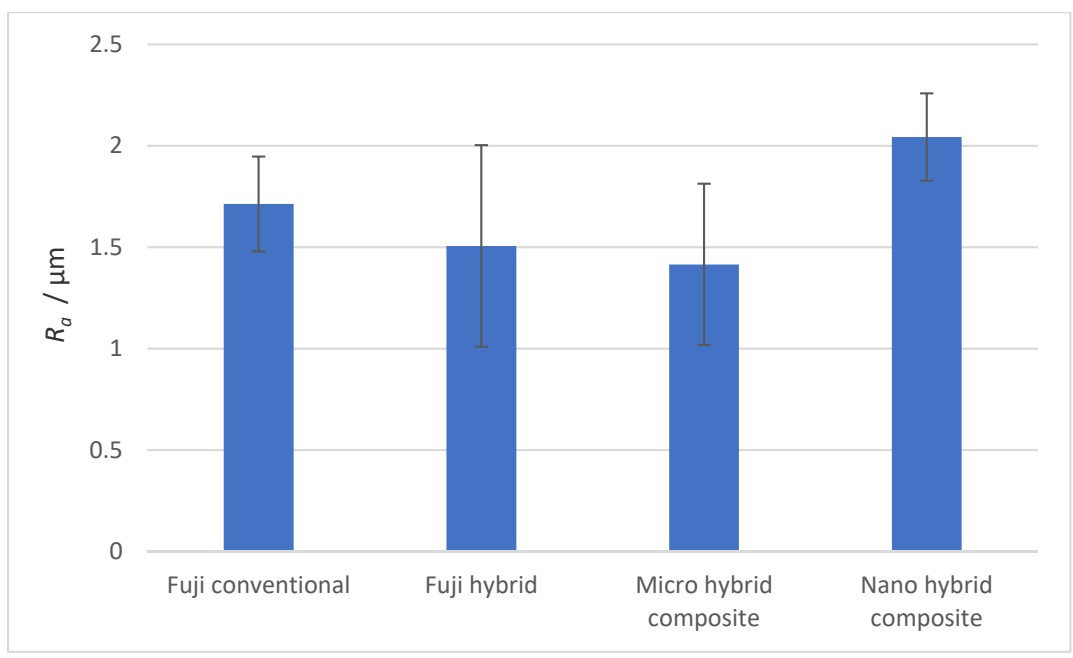

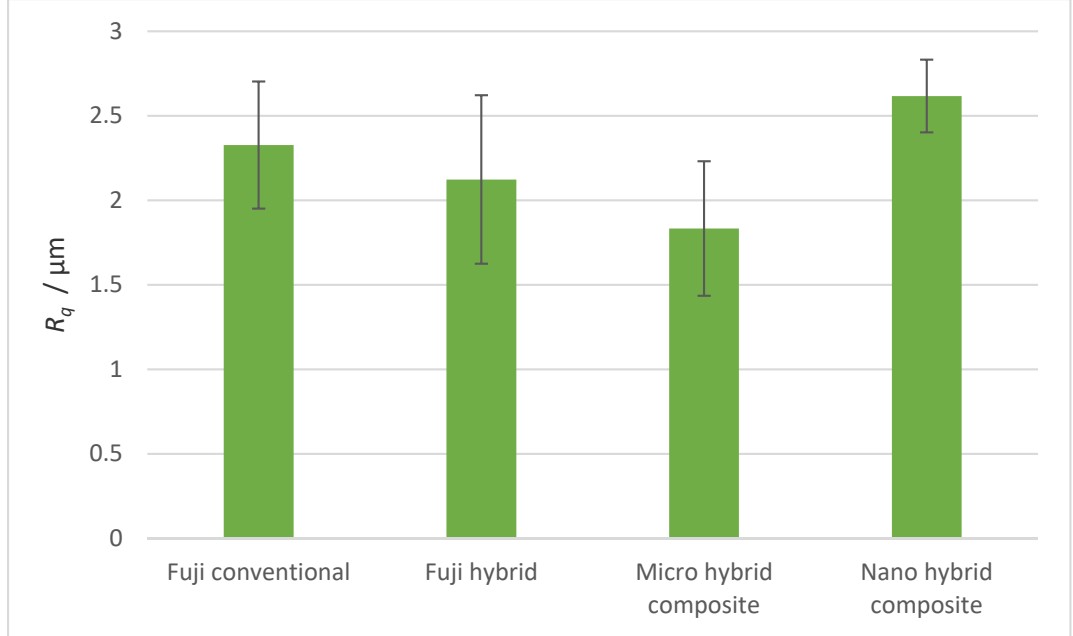

**Figure 1.** Arithmetic average ($R_a$) and root mean square roughness ($R_q$) of samples measured with the profilometer.

### 3.3. Contact Angle

The hydrophobicity of material surfaces is described by the contact angle measurements. Figure 2 presents contact angles of the liquid droplet on the surface of all four materials. Fuji conventional is a hydrophobic material, whereas the Micro hybrid composite and Nano hybrid composite are hydrophilic. The Fuji hybrid's contact angle was 90°.

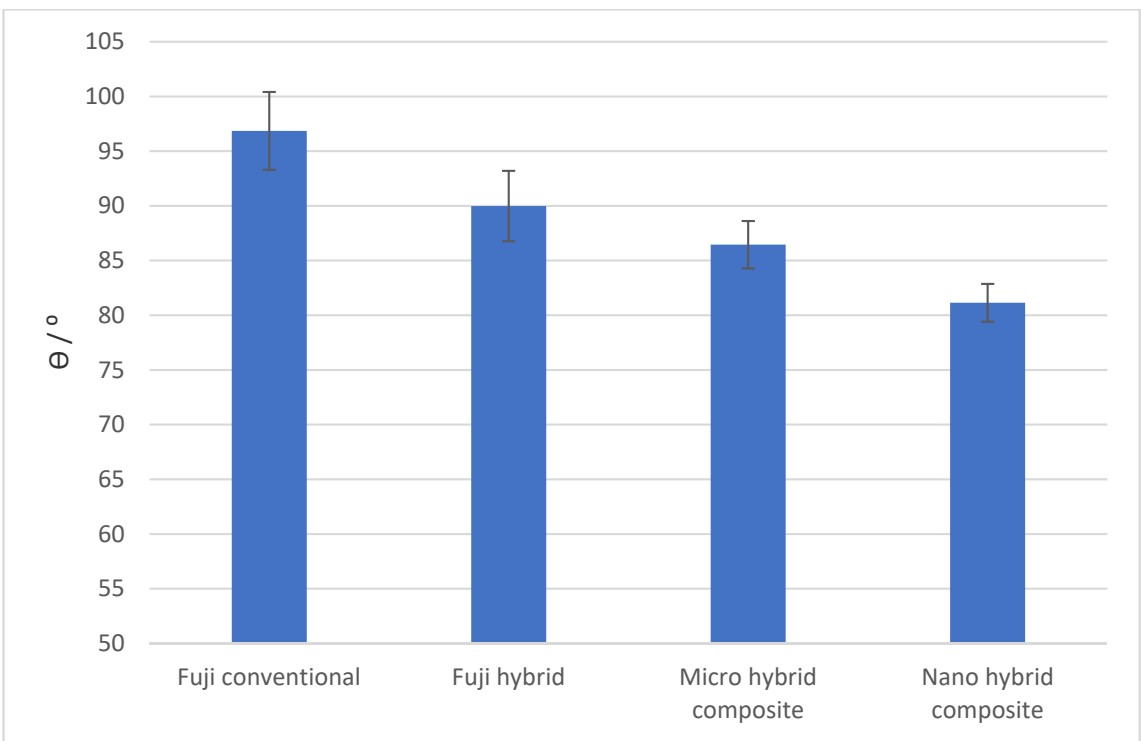

**Figure 2.** Contact angles of a distilled water drop on different dental filling material samples measured with a tensiometer.

### 3.4. Zeta Potential

Table 2 presents zeta potentials of the four dental material surfaces. The results show that all materials are negatively charged. The absolute values of the zeta potential are very similar.

**Table 2.** Zeta potentials of dental materials.

| Material | Zeta Potential (mV) |
| --- | --- |
| Fuji conventional | $-21.1 \pm 0.6$ |
| Fuji hybrid | $-20.9 \pm 0.8$ |
| Micro hybrid composite | $-21.9 \pm 1.6$ |
| Nano hybrid composite | $-22.2 \pm 2.5$ |

### 3.5. XRD and EDS Analysis

The composition of each GIC is described in detail in Table 1 of the article. To confirm the data obtained from the manufacturers, XRD and EDS were performed on all samples. Analysis for Fuji conventional (A, Fuji IX) and Fuji hybrid (B, EQUIA Forte Fil) can be seen in Figure 3. Both samples are a mixture of amorpheous alumino-fluoro-silicate glass with polymer powder. EDS data confirm the chemical composition (with strong Au content due to the application of a conductive coating), but no conclusive information could be obtained from the XRD spectra (after signal processing and background removal).

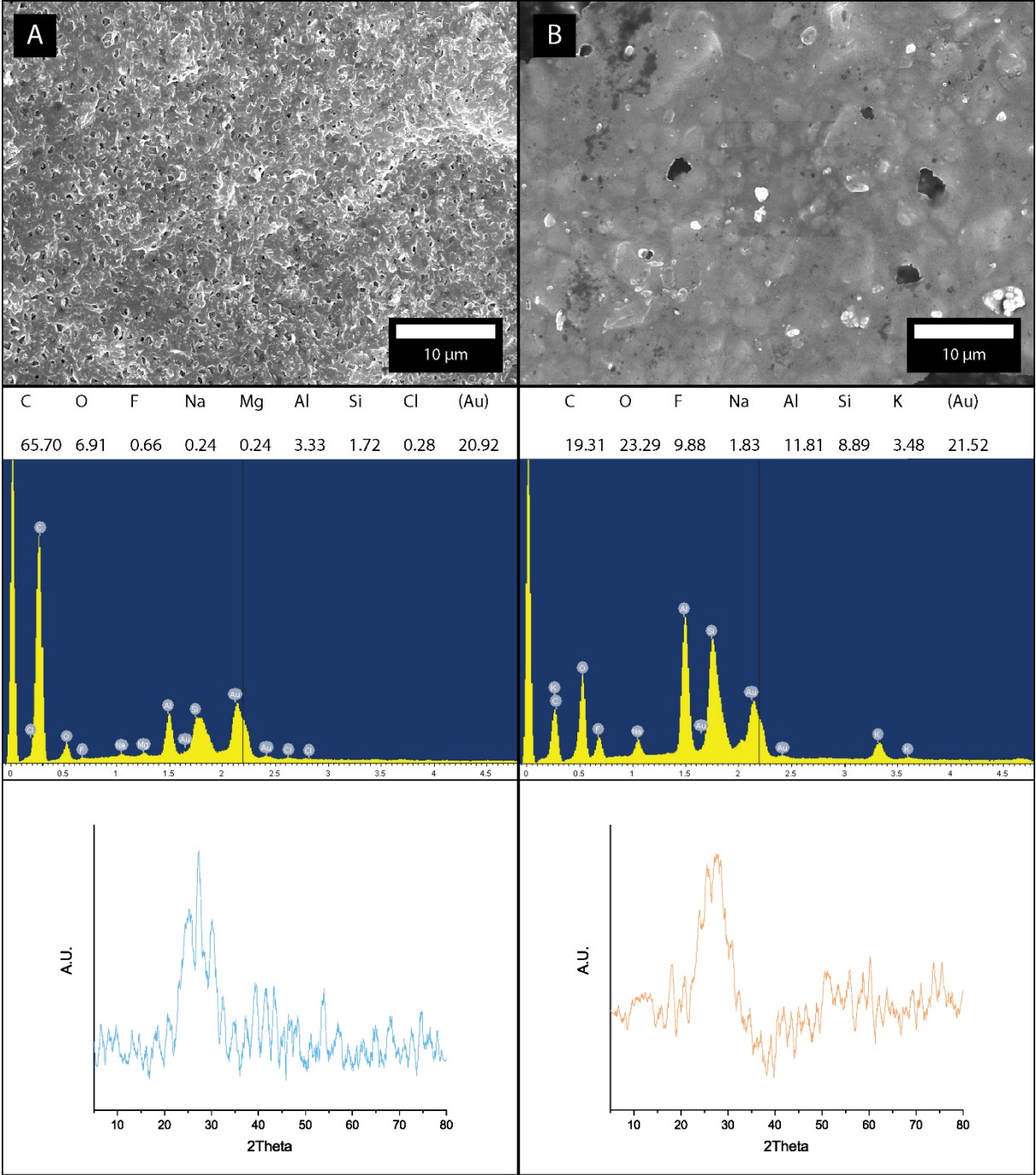

**Figure 3.** Micrographs with EDS and XRD data for: (**A**) Fuji conventional (Fuji IX) and (**B**) Fuji hybrid (EQUIA Forte Fil).

Micro hybrid composite (C, TE Econom) and Nano hybrid composite (D, Tetric Evo Ceram) were both not fully crystalized due to amorphous organic content, but several crystalline phases have been identified using XRD (Figure 4). The majority of peaks can be attributed to orthorhombic ytterbium trifluoride (YbF3—ICSD 00-049-1805). The only noticeable difference between sample C and D for this phase was the intensity of the 002 peak at 40.5°, which was more prominent at sample C. According to the manufacturer's specification and confirmed on the EDS spectra from samples A and B, a silica and alumina mixed oxide is present. XRD confirmed a presence of monoclinic gibbsite (Al(OH)$_3$—ICSD 00-007-0324) with strong (002) peak at 18.28°. The identification of crystalline hexagonal quartz (SiO$_2$—ICSD 00-011-0252) has proven to be difficult due to a high background from

the glass phase and the overlapping of $SiO_2$ (101) peak at 26.19° with the YbF3 (020) peak 26.26°. Increased barium (Br) content is attributed to the barium glass phase in samples C and D.

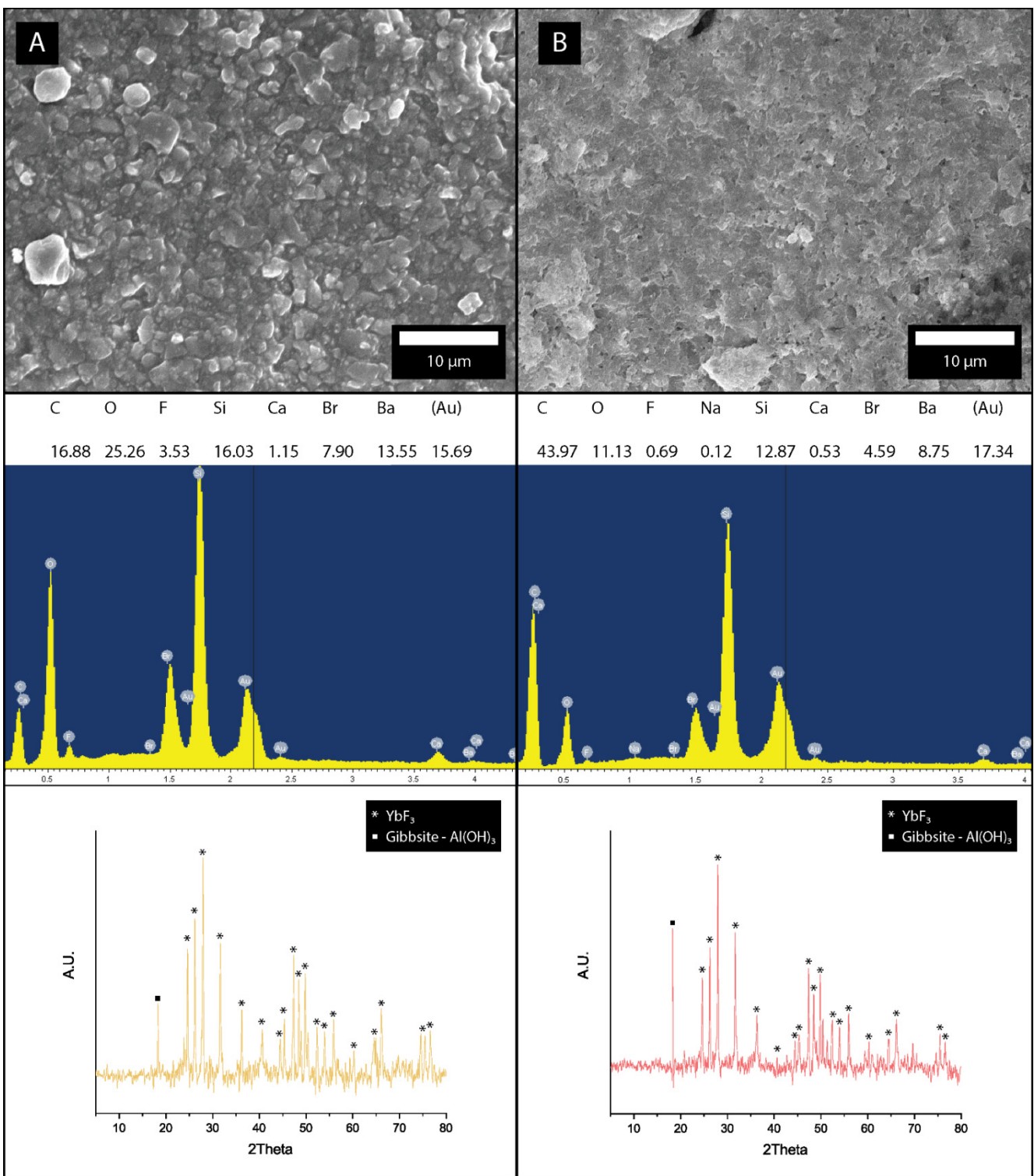

**Figure 4.** Micrographs with EDS and XRD data for: (**A**) Micro hybrid composite (TE Econom) and (**B**) Nano hybrid composite (Tetric Evo Ceram).

### 3.6. Color Parameters

Figure 5 shows hue angles of all four dental surfaces. The hue angle of 90° represents yellow color, while lower values refer to redder shades. A clear difference can be observed between the Fuji conventional, Micro hybrid composite and Nano hybrid composite on one side, and Fuji hybrid on the other side, characterized by a more reddish hue.

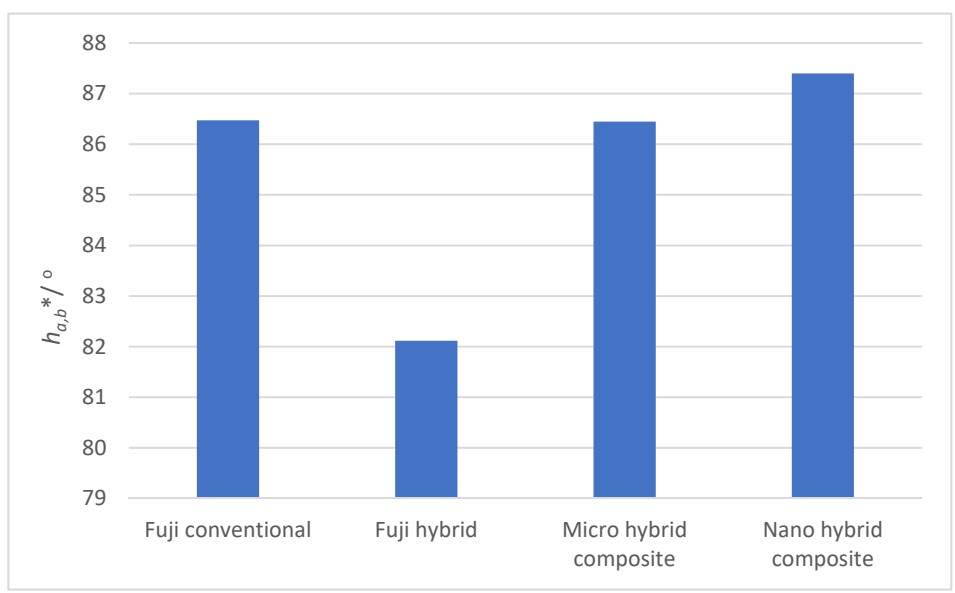

**Figure 5.** Hue angles of four studied dental surfaces.

Figure 6 shows the Chroma ($C^*$) of four dental surfaces. Lower $C^*$ values are associated with lower color tonality or a more intensive grey color. The Micro hybrid composite surface also had a higher $C^*$ value and was obviously more intensively colored as compared to the Fuji conventional, Fuji hybrid and Nano hybrid composite. The Fuji conventional, Fuji hybrid and Nano hybrid composite had more intensive grey notes and fewer yellow and red notes.

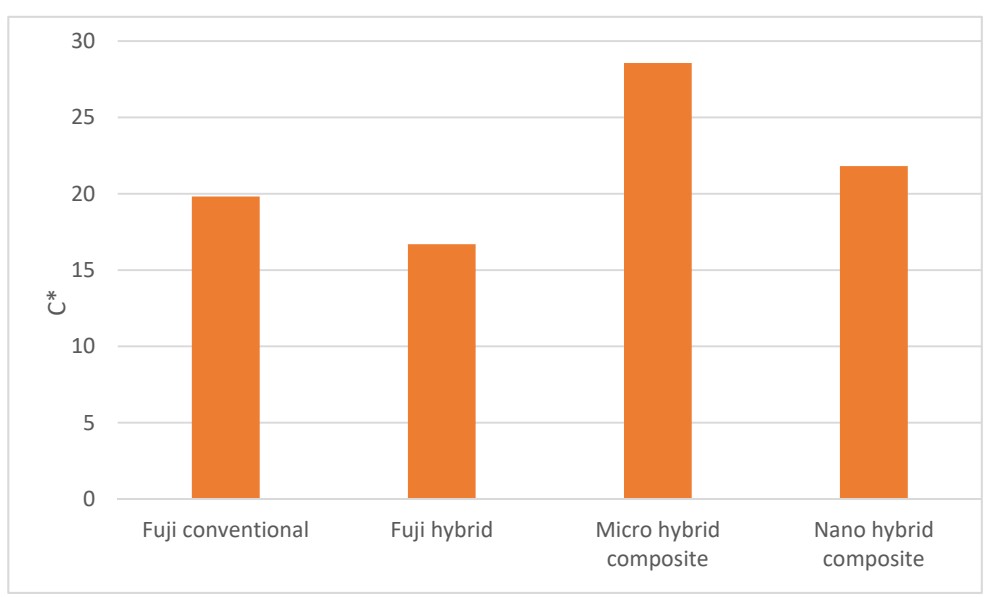

**Figure 6.** Chroma of the studied dental surfaces.

As seen in Figure 7, differences between dental restorative materials Fuji conventional and Fuji hybrid on one side, and Micro hybrid and Nano hybrid composite on the other side, are observed. *L\** axis range from 0 to 100, *L\** = 0 represent black or total absorption, *L\** = 100 represent white or total reflectance. The Micro hybrid composite and Nano hybrid composite are slightly darker as compared to Fuji conventional and Fuji hybrid.

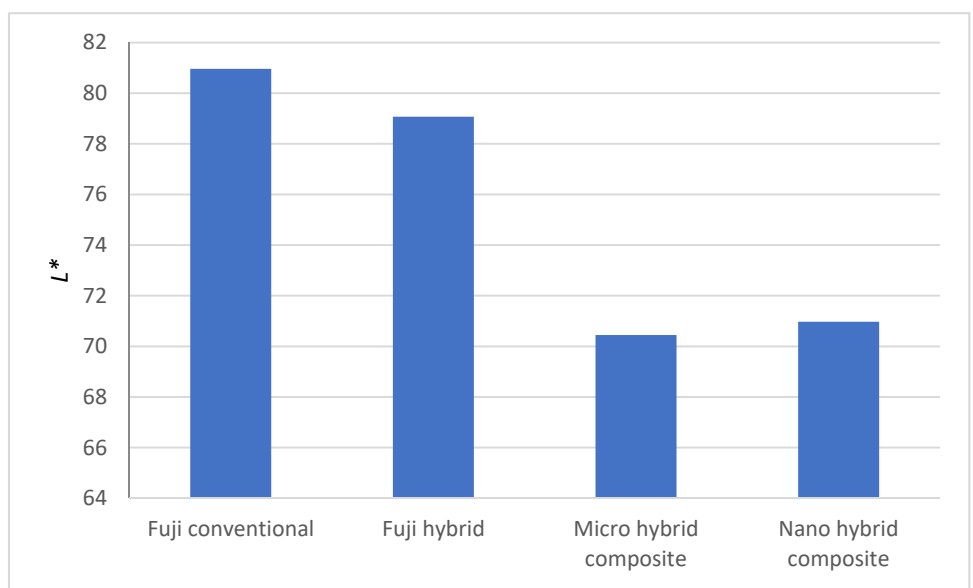

**Figure 7.** Measured *L\** for the samples.

Figure 8 shows the *a\** and *b\** of four dental surfaces. Higher *a\** values correspond to red color, while −*a\** direction depicts a shift toward green color. Fuji hybrid has more intensive red color, followed by the Micro hybrid composite, while Fuji conventional and Nano hybrid composite have similar *a\** values. Regarding *b\** value, +*b* movement represents a shift toward yellow, and −*b\** depicts a shift toward blue color. Slight differences were observed for *b\** value, the exception being Micro hybrid composite that was more intensively yellow and Fuji hybrid that was less intensively yellow.

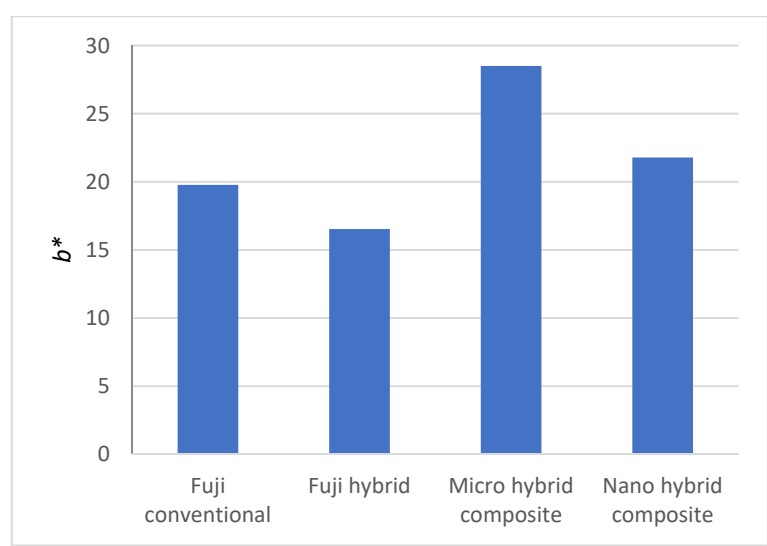

**Figure 8.** Measured *a\** and *b\** for the samples.

### 3.7. Bacterial Adhesion Rate Measurement

The results of bacterial adhesion extent of *S. mutans* on the Fuji conventional, Fuji hybrid, Micro hybrid composite and Nano hybrid composite surfaces is presented in Table 3. The bacterial adhesion extent is given per surface area of 450 μm². The preferential adhesion surface for the tested bacterial strain was the Nano hybrid composite. The lowest bacterial adhesion extent was measured on the Fuji conventional and Fuji hybrid surfaces.

**Table 3.** Bacterial adhesion extent on different material surfaces (number per surface area).

| Material | Bacterial Adhesion Extent/450 μm² |
|---|---|
| Fuji conventional | 5 ± 1 |
| Fuji hybrid | 5 ± 1 |
| Micro hybrid composite | 4.33 ± 1 |
| Nano hybrid composite | 290 ± 10 |

### 3.8. SEM Micrographs

Figure 9 shows SEM micrographs with an adhered bacterial strain of *S. mutans* for Fuji conventional, Fuji hybrid, Micro hybrid composite and Nano hybrid composite surfaces.

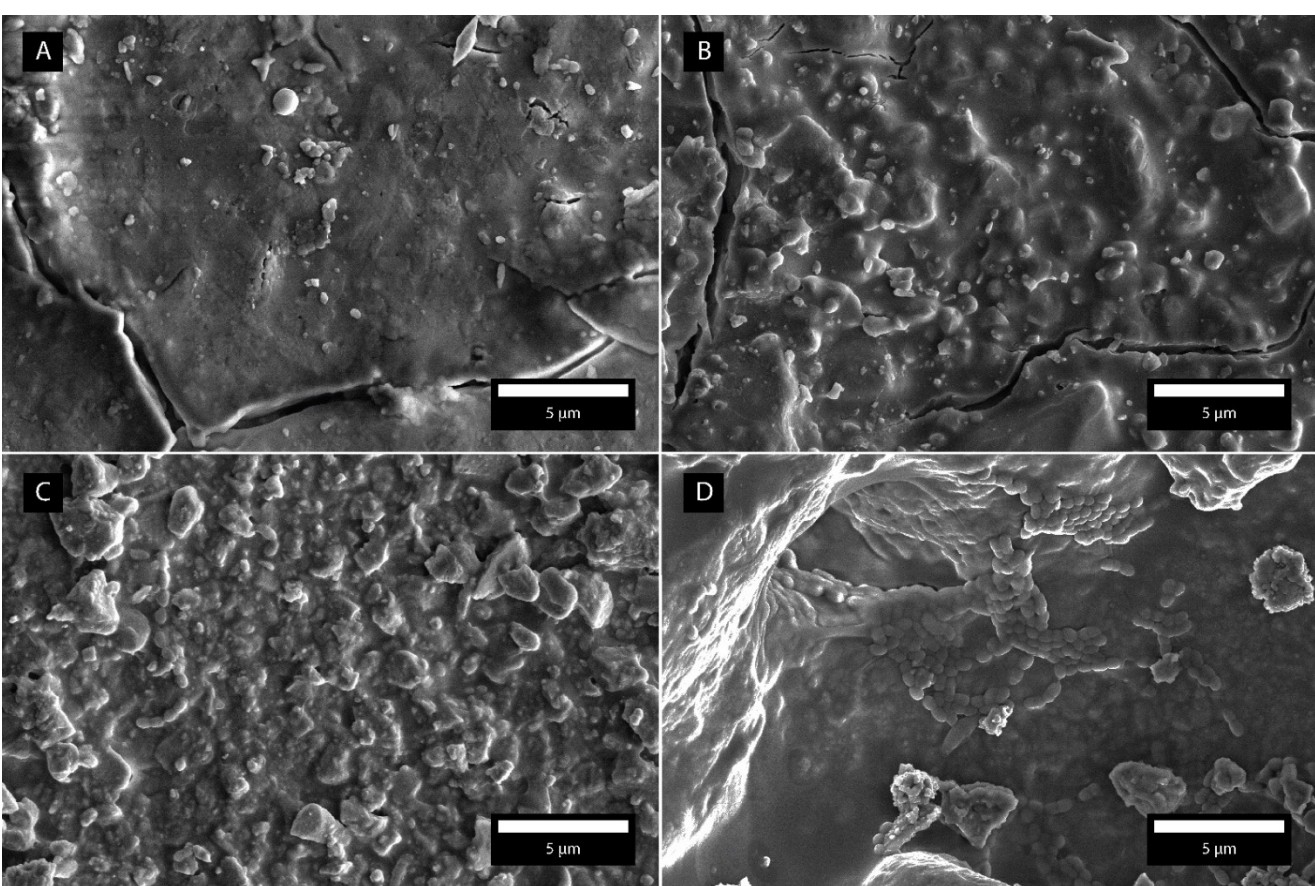

**Figure 9.** SEM micrographs with adhered bacteria *S. mutans* for all four dental surfaces. (**A**) Fuji conventional. (**B**) Fuji hybrid. (**C**) Micro hybrid composite. (**D**) Nano hybrid composite.

### 3.9. Influence of Different Substances (Wine, Black Tea)

In Table 4, we present the results of color parameters for two materials (Fuji Hybrid and Fuji conventional) immersed in wine and black tea. Wine immersion provoked a darker plate color (lower $L^*$ value), more intensive red color (higher $a^*$ value), and a less intensive yellow color (lower $b^*$ value). Black tea immersion resulted in a lighter color

(higher *L** value), more intensive red color (higher a* value), and more intensive yellow color (higher *b**value). Regarding total color difference (Δ*E*), a "non-distinct change" was observed for wine (Δ*E* = 0.9) and a "very distinct" change was observed for black tea (Δ*E* = 3.5). Color changes of the plate submerged in wine or black tea suggest that both substrates adhered to the plate surface, and according to the Δ*E* values, more black tea adhered compared to wine.

**Table 4.** Color parameters for materials immersed in wine and black tea.

| | | **Fuji Conventional** | | | | **Fuji Hybrid** | | |
|---|---|---|---|---|---|---|---|---|
| **Substrate** | **Repetition** | **1** | **2** | **3** | **Repetition** | **1** | **2** | **3** |
| No | *L** | 78.05 | 77.72 | 77.80 | *L** | 79.71 | 79.75 | 79.70 |
| | *a** | 0.67 | 0.67 | 0.64 | *a** | 0.25 | 0.22 | 0.21 |
| | *b** | 14.1 | 14.24 | 14.13 | *b** | 12.82 | 12.80 | 12.76 |
| Wine | *L** | 77.05 | 76.92 | 77.22 | *L** | 75.17 | 75.43 | 75.69 |
| | *a** | 0.78 | 0.79 | 0.76 | *a** | 2.42 | 2.44 | 2.39 |
| | *b** | 13.6 | 13.75 | 13.77 | *b** | 12.03 | 11.97 | 12.06 |
| | (Δ*E*) | | 0.92 | | (Δ*E*) | | 3.93 | |
| Black tea | *L** | 79.20 | 79.84 | 79.09 | *L** | 80.53 | 80.78 | 80.96 |
| | *a** | 1.84 | 1.8 | 1.85 | *a** | 1.57 | 1.52 | 1.45 |
| | *b** | 16.49 | 14.44 | 16.55 | *b** | 15.68 | 16.12 | 16.3 |
| | (Δ*E*) | | 3.48 | | (Δ*E*) | | 5.92 | |

## 4. Discussion

Dental surfaces are often a source of cross-contamination with pathogens. Fuji conventional, Fuji hybrid, Micro hybrid composite and Nano hybrid composite surfaces are currently the most common restorative materials for fillings in teeth. These restorative materials were used in this study to investigate the bacterial adhesion extent. To understand the extent of bacterial adhesion, the surface properties of roughness, contact angle, CIE color parameters, and zeta potential were investigated. In the following paragraphs, we discuss the obtained results.

Glass–ionomer cements (GICs), introduced by Wilson and Kent in 1972, are a special group of dental materials that have a thermal expansion like enamel and are biocompatible with a low toxicity [26]. They also possess a fluoride-releasing property. However, traditional GICs have some drawbacks, such as low fracture toughness, higher occlusal wear, and the need to be protected from initial dehydration and moisture uptake at the early maturation stage compared to other restorative materials, such as amalgam and modern resin composite restorative materials [27–30]. In order to improve the mechanical properties of conventional GICs, resin-modified GICs and the materials containing glass hybrid (EQUIA Forte) were developed.

The composite materials are available with a variety of filler types that affect both their handling characteristics and physical properties [31]. The fillers in composite resins have changed from macroparticles to nanoparticles, through which aesthetic and mechanical properties of the materials have been improved [32,33]. Besides the traditional filler particles, most Nanohybrid resin composites contain small concentrations of nanofillers and/or nanofiller clusters that increase the filler load, improve mechanical properties, and produce highly polishable surfaces [33,34].

The exact information about material properties of the studied samples is very important in such types of studies. Therefore, we performed physico-chemical characterizations of the studied materials by means of XRD and EDS measurements, as presented in Figures 3 and 4. The obtained results (primarily EDS micrographs) confirmed the chemical composition stated by the producers. Among different surface properties, the surface roughness, charge, and hydrophobicity play very important roles in the bacterial adhesion process [35–37]. The roughness measurements show that the arithmetic surface roughness

lies between 1.4 μm and 2.1 μm, with the Nano hybrid composite having the highest and Micro hybrid composite the lowest roughness. The Nano hybrid composite had roughness $R_a = (2 \pm 0.2)$ μm and $R_q = (2.6 \pm 0.4)$ μm. Increased roughness might be a predisposing factor to microbial colonization. We found that the highest roughness was measured for our Nano hybrid composite surfaces, whereas glass–ionomer materials had slightly lower roughness.

The hydrophobicity of studied materials is found to be more diverse. The Nano hybrid composite surfaces are hydrophilic (with the contact angle close to 80°), as was the Micro hybrid composite surface with a contact angle of 86°. The Fuji hybrid surface had a contact angle of 90°. The conventional hybrid surface was hydrophobic with a contact angle of 97°. All materials were negatively charged, with very similar zeta potentials in the region between $20.9 \pm 0.8$ mV and $22.2 \pm 2.5$ mV.

The bacterial adhesion measurements showed that the greatest adhesion extent was obtained for the Nano hybrid composite surface, which is hydrophilic and possesses a significantly greater risk of bacterial adhesion and the corresponding biofilm formation [38]. An additional factor for pronounced bacterial adhesion is the higher roughness of the Nano hybrid composite compared to other materials. The impact of roughness is in accordance with our previous findings, where we showed that the increased surface roughness causes higher bacterial adhesion [39]. In the study of Poggio et al. [40], composite surfaces were prepared with a higher roughness than glass–ionomer cements, and correspondingly, the bacterial adhesion showed opposite trends compared to our study.

Among the glass–ionomer cements and Micro/Nano hybrid composite dental surfaces studied here, Fuji hybrid differentiated from others by having a lower hue angle and a higher *a** value, all associated with a more intensive red color. Fuji hybrid and Fuji conventional are also lighter in color, as evidenced by a higher *L** value as compared to the Micro hybrid and Nano hybrid composites. Micro hybrid composite differentiated from others by having the highest *C** color parameter, associated with saturated, more vivid colors, while the Fuji hybrid had a more muted color and a less yellow color parameter.

The problems associated with aesthetic restorations are those related to color matching procedures. Considering the unlimited possibilities of available shades and opacities used for reproducing the optical properties of the dental structures, the initial outcome of a direct restoration may be excellent [41]. Composites have versatility and can be provided in different shades and opacities, similar with optical properties of natural dental structures. Currently, glass–ionomers are preferred for reproducing the optical properties due to better properties. The color of most human teeth corresponds to a small range of the color space from yellowish-white to light-brown, and the degradation and aging of teeth is usually associated with yellowness. The color change of composite fillings in different colored media over time is a common problem in cosmetic dentistry, which causes the need to replace the fillings. Matching the colors of the dental material with the color of the oral tissue is one of the most important characteristics of restorative materials. Color stability through the period of restoration functionality directly determines the longevity of the restoration.

Today, in the era of highly aesthetic dentistry, patients require aesthetically satisfactory restorations, which retain their initial color and appearance for a longer period. Therefore, proper color matching with the surrounding tooth tissue is important not only in the first period of function, but also over a longer period.

Dentine has fluorescence excitation peaks at wavelengths of 300, 325, 380 and 410 nm, with corresponding emission maxima at ca. 350, 400, 450 and 520 nm [42]. Enamel excitation peaks have been found at 285 and 330 nm, and emission maxima at 360 and 410 nm [43].

Hasegawa et al. [44] measured the color in five different locations along the tooth axis of the labial surface of the central incisors using spectrophotometry, and found significant variations in the lightness *L**, red-green values *a**, and yellow-blue values *b**. The translucency of natural teeth was also shown to decrease from the incisal site towards the central site. The cervical regions have been shown to have the lowest translucency.

Food and beverages adhere to some degree to dental surfaces and impact the adhesion and growth of microorganisms. From that point of view, compounds with antimicrobial activity are of special interest. Phenolic compounds are present in relatively high amounts in red wine and black tea. Both beverages are also consumed quite frequently; therefore, we tested their adhesion on both Fuji composites through changes in color parameters.

Immersion of both glass–ionomer cements in red wine or black tea resulted in adhesion of both beverages on their surface. As a result of immersion in wine, *L\** and *b\** values decreased, while the *a\** value increased; cements thus became darker and had a more intensive red and less intensive yellow color. Comparing both glass–ionomer cements, more profound color changes were observed for Fuji hybrid, suggesting that wine constituents adhered to this glass–ionomer cement more intensively. This is also reflected by the color difference Δ*E* parameter, which amounted to 0.92 (Fuji conventional) and 3.93 (Fuji hybrid). In their study, Chakravarthy and Clarence [45] showed that the restorative GIC had more discoloration than the composite. Additionally, Hse et al. [46] showed that the GICs lack color stability due to the polyacid content of the material which can be explained by the degradation of metal polyacrylate salts.

Contrary to immersion in wine, black tea immersion resulted in an increase in *L\** and *b\** values. Comparing wine and black tea, a more profound increase in the *a\** color parameter was observed for black tea. Summarizing the color changes of studied glass–ionomer cements, black tea immersion provoked more intensive red and yellow color components as compared to wine. Again, more pronounced changes of color parameters were observed for Fuji hybrid compared to Fuji conventional. Bearing in mind color difference, greater Δ*E* was observed for black tea (3.48 vs. 0.92 for wine) in the case of the Fuji hybrid, and 5.92 vs. 3.93 in the case of Fuji conventional, respectively. Indeed, Δ*E* values correspond to very distinct changes and thus demonstrate that Fuji hybrid allows more intensive adhesion for wine and black tea which is related to quite high surface roughness of 1.5 μm. Lee et al. [47] concluded that staining ability was influenced by each composite monomer and filler composition. The study of Yildiz et al. included composites as well as glass–ionomer cements, and showed that composites have the lowest Δ*E* values [48].

We also examined the possible inhibitory effects of black tea and red wine on the growth of *S. aureus* and *S. mutans*. The measurements were performed by a standard diffusion method for the antibiogram with Mueller–Hinton medium. Only black tea showed a slight inhibition zone, while the other tested ingredients of wine did not inhibit the growth of these two types of bacteria.

## 5. Conclusions

In this study, we examined the impact of two glass–ionomer cements (Fuji conventional and Fuji hybrid) and two ceramic composites on bacterial adhesion. The surface topography, roughness, hydrophobicity, and zeta potential were measured, and from the SEM micrographs the bacterial adhesion extent was determined. We showed that the highest bacterial adhesion was on the Nano hybrid composite surface, which, in this study, was associated to a low contact angle, higher roughness, and the most negative zeta potential. This study helps in understanding which dental surfaces reduce bacterial adhesion when exposed to the oral environment. Food rich in antioxidants such as black tea might modulate the growth and adhesion of bacteria.

We also took CIE color parameters of dental surfaces under consideration. The influence of two quite frequently consumed beverages (red wine and black tea) on color parameter changes was examined. Black tea adhered more intensively than wine upon dipping dental surfaces in both beverages, and also showed a higher inhibitory effect on bacterial growth. In the possible continuation of this study, more in-depth studies of the influence of different food on surface characteristics should be considered. Moreover, studies encompassing the adhesion of microorganisms to food-modified dental restorations would enlighten the interactions between microorganisms, food, and dental restorations.

**Author Contributions:** Conceptualization, K.B., O.D. and R.V.; data curation, A.Z. and A.A.; formal analysis, A.A.; investigation, K.B., E.T., A.Z., O.D., M.O. and A.A.; supervision, K.B.; writing—original draft, K.B., O.D. and D.K.; writing—review & editing, D.K., Ž.K., M.O., V.V. and O.D. All authors have read and agreed to the published version of the manuscript.

**Funding:** K.B., D.K., Ž.K. and O.D. thank the COST action CA 15216-ENBA for the financial support. K.B., O.D., Ž.K., M.O. and V.V. thank the bilateral project between Slovenia and BiH. K.B. thanks ARRS through the program "Mechanisms of health maintenance for financial support".

**Institutional Review Board Statement:** Not applicable.

**Informed Consent Statement:** Not applicable.

**Data Availability Statement:** Data is contained within the article.

**Acknowledgments:** E.T. thanks CEEPUS for the grant CIII-HR-1108-02-1819-M-130417 in the frame of the network "Colloids and nanomaterials in education and research".

**Conflicts of Interest:** The authors declare no conflict of interest.

**Abbreviations**

| | |
|---|---|
| CIE | Commission Internationale de l'Eclairage |
| $\Delta E$ | color difference |
| SEM | scanning electron microscope |
| BHI | brain–heart infusion |
| IX | Fuji conventional |
| Fil | Fuji hybrid EQUIA Forte |
| TE Econom | Micro hybrid composite |
| Tetric Evo Ceram | Nano hybrid composite |
| GIC | Glass–ionomer cement |
| $R_a$ | arithmetic average roughness |
| $R_q$ | root mean square roughness |
| $\zeta$ | zeta potential |

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
