# Peer review of "Bacterial Adhesion on Glass–Ionomer Cements and Micro/Nano Hybrid Composite Dental Surfaces"

_coatings, doi:10.3390/coatings11020235_

Round 1

Reviewer 1 Report

The work of the researchers is well conducted, but I want to give them some suggestions:

  • The introduction is too much longer (the discussion is shorter than the introduction). I suggest to modify and shift in the discussion most of information contained in lines 52-87. These information should be resumed for the introduction.
  • Section 2 (materials and methods) is adequately described.
  • Section 3 (results) are clearly presented. I appreciated the subdivision into subparagraphs for the different parameters included in the investigation.
  • Section 4 (discussion) may be modified as previously suggested.
  • Section 5 (conclusions) is supported by the results. However, in my opinion, it is not necessary the repetition in lines (386-391, 395-396). These information are more appropriate for the discussion. The conclusions should be reformulated.

Author Response

Reviewer 1

Open Review

Comments and Suggestions for Authors

The work of the researchers is well conducted, but I want to give them some suggestions:

Our reply: We thank the Reviewer #1 for the supportive comments, and we have tried to satisfactorily address the modifications requested, as detailed below.

The introduction is too much longer (the discussion is shorter than the introduction). I suggest to modify and shift in the discussion most of information contained in lines 52-87. These information should be resumed for the introduction.

Our reply: As suggested by the reviewer we have shortened the introduction. Most of the lines 52-87 were shifted to the Discussion. The remaining lines in the introduction are marked red (lines 51-67). New parts in the Discussion are marked red (lines 333-359, 371-375,385).

Section 2 (materials and methods) is adequately described.

Our reply: Thanks

Section 3 (results) are clearly presented. I appreciated the subdivision into subparagraphs for the different parameters included in the investigation.

Our reply: The results were already subdivided for different parameters (roughness, contact angle, zeta potential, microbial adhesion extent).

Section 4 (discussion) may be modified as previously suggested.

Our reply: New parts in the Discussion are marked red (lines 333-359, 371-375,385).

Section 5 (conclusions) is supported by the results. However, in my opinion, it is not necessary the repetition in lines (386-391, 395-396). This information is more appropriate for the discussion. The conclusions should be reformulated.

Our reply: The conclusion was rewritten and reformulated (lines 394-404). The suggested lines were removed.

Reviewer 2 Report

This is my personal review report of the manuscript entitled "Bacterial Adhesion on Glass Ionomer Cements and Micro/Nano Hybrid Composite Dental Surfaces" proposed by K. Bohinc, E. Tintor, D. Kovačević, R. Vidrih, AM. Zore, A. Abram, Z.Kojić, M.Obradović, V. Veselinović and O. Dolić for publication in the journal Coatings.

The research work deals with the characterization of bacterial adhesion on the surface of glass ionomer cements used for dental applications. Four different dental materials are tested.

In addition to bacterial adhesion, several surface characterizations are provided to assess the roughness, the contact angles, the colours, and the zeta potentials of the biomaterial surfaces.

In my opinion, the study is more a technical report of experimental results rather than a real scientific paper. The work contains characterizations of commercial materials with no discussion of the obtained results. The reader would like to know what the specificity of each material is. Why is it interesting to compare the behaviours of these samples? Why these results are interesting in comparison with the literature? What is the novelty of the work? Any new development? Any new perspectives for science?

The presented results are not much different for the four samples. Why is it interesting to compare them? Is there any specific impact of their composition or other material properties? Moreover, more material characterizations are necessary, e.g., XRD, FTIR, XPS, EDS…

For these reasons, I recommend to reject the paper.

Author Response

Reviewer 2

Open Review

Comments and Suggestions for Authors

This is my personal review report of the manuscript entitled "Bacterial Adhesion on Glass Ionomer Cements and Micro/Nano Hybrid Composite Dental Surfaces" proposed by K. Bohinc, E. Tintor, D. Kovačević, R. Vidrih, AM. Zore, A. Abram, Z.Kojić, M.Obradović, V. Veselinović and O. Dolić for publication in the journal Coatings.

The research work deals with the characterization of bacterial adhesion on the surface of glass ionomer cements used for dental applications. Four different dental materials are tested.

 In addition to bacterial adhesion, several surface characterizations are provided to assess the roughness, the contact angles, the colours, and the zeta potentials of the biomaterial surfaces.

 In my opinion, the study is more a technical report of experimental results rather than a real scientific paper. The work contains characterizations of commercial materials with no discussion of the obtained results. The reader would like to know what the specificity of each material is. Why is it interesting to compare the behaviours of these samples? Why these results are interesting in comparison with the literature? What is the novelty of the work? Any new development? Any new perspectives for science?

The presented results are not much different for the four samples. Why is it interesting to compare them? Is there any specific impact of their composition or other material properties? Moreover, more material characterizations are necessary, e.g., XRD, FTIR, XPS, EDS… 

For these reasons, I recommend to reject the paper.

Our reply: We are sorry that the Reviewer #2 felt that our manuscript has no novelty, no new development and no new perspectives for science. We have thus done our best to integrate this worry into the overall concept of the revision of our manuscript. We would note here that based also on the comments of Reviewer #1 and #3 (who appeared to be not quite so disappointed with the overall quality of our manuscript), we have improved things by analyzing some proposed material characterizations.

Composition of each GIC is described in detail in Table 1 of the article (line 167). Doing a separate analysis of chemical composition is, in our opinion, outside the scope of this article. The exact identification of correct phases would also be challenging due to the composite nature of the GICs (amorphous vs. crystalline, glass/ceramics vs. polymers). We did, however, provide the additional EDS data on each of the samples in the supplement materials. We omitted other methods because they will provide very limited information (either only the information of crystalized phases for XRD, or surface-limited information in the case of XPS).

Nevertheless, we also want to stress that our research consider dental surface mainly used in nowadays dental medicine. Our main purpose was to find out the predominantly used dental material with lower bacterial adhesion and consequently lower secondary caries formation.

Reviewer 3 Report

The work describes bacterial adhesion on cements and micro/nano hybrid composite dental surfaces. The authors presented various results to compare the nature of bacterial adhesion on the surfaces. I suggest the authors make some improvements before publications. Below are my suggestions.

  1. The introduction is a bit long and is beyond the word count usually recommended for an introduction in an article. I suggest the authors condense it by avoiding redundancy. For instance, the authors should consider merging paragraphs dealing with colouration between lines 51 till 87 and make a single paragraph.
  2. I also recommend the authors to use references in order of their use in the text. To clarify my point references 33-38 were included in the introduction might be added after the whole text is written. So use of endnote/or manual ordering of references can be used.
  3. In the result section the authors tried to give details that are worth mentioning in the discussion section. I recommend moving the two paragraphs between lines 151-166 to discussion section.
  4. List of abbreviation at the end of the text
  5. I also recommend the merging fingers/ graphs, it is possible to put figures 1 and to 2 as figure 1, both deal with roughness. It is a bit surprising the huge variation within the samples used as seen in the error bar, how could it be explained?
  6. In Figure 8, I suggest the authors replace figure c with a figure in 10µm scale to help readers compare the difference among the surfaces. 

Author Response

Reviewer 3

Open Review

Comments and Suggestions for Authors

The work describes bacterial adhesion on cements and micro/nano hybrid composite dental surfaces. The authors presented various results to compare the nature of bacterial adhesion on the surfaces. I suggest the authors make some improvements before publications. Below are my suggestions.

Our reply: We thank the Reviewer for the recognition of the value of our study, and for the time and effort that was obviously put in here. At this level, we believe we have integrated all of the comments of Reviewer #3 listed below into this revision.

The introduction is a bit long and is beyond the word count usually recommended for an introduction in an article. I suggest the authors condense it by avoiding redundancy. For instance, the authors should consider merging paragraphs dealing with colouration between lines 51 till 87 and make a single paragraph.

Our reply: As suggested by the reviewer we have shortened the introduction. Most of the lines 52-87 were shifted to the Discussion. The remaining lines in the introduction are marked red (lines 51-67). New parts in the Discussion are marked red (lines 338-364, 376-390,393).

I also recommend the authors to use references in order of their use in the text. To clarify my point references 33-38 were included in the introduction might be added after the whole text is written. So use of endnote/or manual ordering of references can be used.

Our reply: Thank you for this suggestion. The references have been reordered in accordance with the journal’s rule.

In the result section the authors tried to give details that are worth mentioning in the discussion section. I recommend moving the two paragraphs between lines 151-166 to discussion section.

Our reply: The lines 151-166 were moved to the Discussion section (now lines 338-364). New parts in the Discussion are marked in red (lines 376-390, 376-380,390-393).

List of abbreviation at the end of the text

Our reply: List of abbreviations is now prepared (pages 14-15).

I also recommend the merging fingers/ graphs, it is possible to put figures 1 and to 2 as figure 1, both deal with roughness. It is a bit surprising the huge variation within the samples used as seen in the error bar, how could it be explained?

Our reply: Figures 1 and 2 were merged. We agree with the reviewer that in Figures 1 and 2 the error of the roughness is large (presented as standard deviation). We check the errors and found out that the errors were overestimated. We recalculated the error in accordance with our previous studies (Bohinc K et al, Metal surface characteristics dictate bacterial adhesion capacity. Int J Adhes Adhes 2016; 68: 39-46.) and show the standard error.

For each material we made three roughness measurements. According to the previous studies this should be sufficient (Farkas et al, Measurement uncertainty of surface roughness measurement, Materials Science and Engineering, 448, 2018, 012020).

Additional comments about the influence of roughness were made in lines 313-315 and 328-330.

In Figure 8, I suggest the authors replace figure c with a figure in 10µm scale to help readers compare the difference among the surfaces. 

Our reply: Fig. 8c has been replaced by a figure with 10µm scale.

Round 2

Reviewer 2 Report

I have reviewed the corrected version of the paper coatings-1091132  entitled 'Bacterial Adhesion on Glass Ionomer Cements and Micro/Nano Hybrid Composite Dental Surfaces '.

First of all, I have to mention that in my first report, I did not consider that the paper has no novelty but that the novelty of the paper must be highlighted more explicitly for the reader.

I have to admit that this corrected version contains more discussion, providing more explanations about the observed results supported by numerous references on similar topics. The improvement of the discussion appropriately corrects one of the points I have mentioned in my previous comments.

However, even if the biological results are intersting, even if the discussion about the surface characterization is better, I still consider that the paper needs more physic-chemical characterizations results , e.g. XRD, EDS, XPS... This is my personal point of view regarding the scope of the journal Coatings.

According to this, I recommend now a major revision including such results to match more with the scope of the journal.

Author Response

Reviewer 2

First of all, I have to mention that in my first report, I did not consider that the paper has no novelty but that the novelty of the paper must be highlighted more explicitly for the reader.

Our reply: We would like to thank the reviewer.

I have to admit that this corrected version contains more discussion, providing more explanations about the observed results supported by numerous references on similar topics. The improvement of the discussion appropriately corrects one of the points I have mentioned in my previous comments.

However, even if the biological results are intersting, even if the discussion about the surface characterization is better, I still consider that the paper needs more physic-chemical characterizations results , e.g. XRD, EDS, XPS... This is my personal point of view regarding the scope of the journal Coatings.

Our reply: We did our best to provide additional physico-chemical characterization of studied samples. Therefore, we performed suggested XRD and EDS experiments and analyzed the obtained results. The experimental methods are described in the subsections 2.6 and 2.7. The results, presented as two new Figures, are given in subsection 3.5 and discussed in the section Discussion, 4th paragraph. With EDS analysis the main elements in the samples were detected. In our opinion XPS analysis would not give any new insight in the system and therefore the analysis was not performed. We hope that with these additional results our manuscript fits even more into the scope of the journal Coatings.